# COVID-19: Current Status in Gastrointestinal, Hepatic, and Pancreatic Diseases—A Concise Review

**DOI:** 10.3390/tropicalmed7080187

**Published:** 2022-08-16

**Authors:** Jorge Aquino-Matus, Misael Uribe, Norberto Chavez-Tapia

**Affiliations:** Digestive Diseases and Obesity Clinic, Medica Sur Clinic & Foundation, Mexico City 14070, Mexico

**Keywords:** COVID-19, SARS-CoV-2, diarrhea, liver, pancreas, inflammatory bowel diseases, liver transplantation

## Abstract

The gastrointestinal tract plays an important role in the pathogenesis of COVID-19. The angiotensin-converting enzyme 2 receptor and the transmembrane protease serine 2 receptor bind and activate SARS-CoV-2 and are present in high concentrations throughout the gastrointestinal tract. Most patients present with gastrointestinal symptoms and/or abnormal liver function tests, both of which have been associated with adverse outcomes. The mechanisms of liver damage are currently under investigation, but the damage is usually transient and nonsevere. Liver transplantation is the only definitive treatment for acute liver failure and end-stage liver disease, and unfortunately, because of the need for ventilators during the COVID-19 pandemic, most liver transplant programs have been suspended. Patients with gastrointestinal autoimmune diseases require close follow-up and may need modification in immunosuppression. Acute pancreatitis is a rare manifestation of COVID-19, but it must be considered in patients with abdominal pain. The gastrointestinal tract, including the liver and the pancreas, has an intimate relationship with COVID-19 that is currently under active investigation.

## 1. Introduction

The new coronavirus SARS-CoV-2 infection (COVID-19) reported in Wuhan, China, in December 2019, and by 11 March 2020, the World Health Organization declared it a global pandemic [1]. Time has passed, many people have succumbed to the infection, and a global effort has been carried out to study the virus, both to develop an effective vaccine and to prevent further complications among survivors. Coronaviruses are 65–125 nm diameter (26–32 kb genome) positive monocatenary RNA viruses from the family *Coronaviridae* and subfamily *Orthocoronavirinae*, which are subdivided into *Alphacoronavirus*, *Betacoronavirus*, *Gammacoronavirus*, and *Deltacoronavirus* [2]. The two previous *Betacoronavirus* pandemics were SARS-CoV in 2003 and MERS-CoV in 2012, with mortality of 10% and 37%, respectively [3]. Interestingly, the new coronavirus SARS-CoV-2 genome shows 96.2% homology with the RaTG13 virus in bats (Bat-RaTG13 and Bat-SL-CoVZXC21) [4].

Coronaviruses are known to cause severe acute respiratory distress syndrome and death, and the gastrointestinal tract has been described as playing a key role in the route of infection, clinical manifestations, and disease outcomes. In addition, many gastrointestinal diseases (e.g., autoimmune hepatitis, Crohn’s disease) require immunosuppressive treatment and whether the clinical course or risk of complications may be halted by treatment for COVID-19 infection is still under investigation. Moreover, liver transplant as the only intervention for terminal liver disease has also been affected by the pandemic. Specifically, most transplant programs in the world have been halted, which will result in longer waiting lists, liver decompensation, and death in the short term for many such patients.

The objective of the current review is to describe the intimate relationship between the gastrointestinal tract, including the liver and pancreas, and the pathogenesis, clinical course, and outcomes of the COVID-19 pandemic.

## 2. The Gastrointestinal Tract in the Pathogenesis of COVID-19

Coronaviruses express four structural proteins: spike (S), membrane (M), envelope (E), and nucleocapsid (N). The S protein receptor-binding domain shares 75% of the amino acid sequence of the SARS-CoV virus [2]. The conformational change of protein S, which binds with the angiotensin-converting enzyme 2 (ACE2) receptor and transmembrane protease serine 2 receptor (TMPRSS2) of the host cell, allows fusion of the viral envelope with the cell membrane and internalization of the virus [5]. Interferon upregulates the expression of ACE2 and TMPRSS2 receptors in the nasal secretory cells, type II pneumocytes, and enterocytes. Thus, the tropism of the virus is determined by the tissue distribution of ACE2 and TMPRSS2 receptors [6,7]. The affinity of SARS-CoV-2 for the ACE2 receptor in the gastrointestinal tract is 10–20 times greater than that of SARS-CoV [8].

Although fecal isolates of SARS-CoV-2 are capable of infecting cells cultured in vitro, fecal–oral transmission, though possible, has not been demonstrated [9]. In a retrospective study, a positive fecal real-time polymerase chain reaction (RT-PCR) was detected two to five days after a positive sputum RT-PCR with fecal excretion of the virus for up to 11 days [10].

Interestingly, SARS-CoV-2 RNA was identified in the sewage water of hospitals in Beijing and when seeded, could remain infectious for 14 days at 4 °C and 2 days at 20 °C [11]. Evidence for gastrointestinal infection of SARS-CoV-2 is controversial. For example, a 78-year-old man with COVID-19 developed gastrointestinal bleeding during hospitalization, with a hematoxylin and eosin (H&E) stain of endoscopic samples showing damage with infiltrating lymphocytes in the esophageal epithelium and plasma cells with interstitial edema in the stomach, duodenum, and rectum [12].

Gut microbiota could play a potential role as a diagnostic or prognostic biomarker in patients with COVID-19, as reported in a comprehensive systematic review including 1668 studies [13]. Patients with COVID-19 develop gut microbiota dysbiosis with depletion of *Ruminococcus*, *Alistipes*, *Eubacterium*, *Bifidobacterium*, *Faecalibacterium*, *Roseburia*, *Fusicathenibacter*, and *Blautia*, and enrichment of *Eggerthella*, *Bacteroides*, *Actinomyces*, *Clostridium*, *Streptococcus*, *Rothia*, and *Collinsella.* A dysregulated gut environment could increase the expression of ACE2 in the gut and favor more severe disease [14]. Table 1 summarizes examples of basic and translational implications of COVID-19.

## 3. COVID-19 and Gastrointestinal Symptoms

Most patients with COVID-19 report nonspecific symptoms, including fever, headache, fatigue, arthralgias, myalgias, and general malaise. Because SARS-CoV-2 targets primarily the respiratory tract, additional symptoms such as cough, dyspnea, and anosmia are also reported [19]. Furthermore, in most published studies the prevalence of gastrointestinal symptoms varies between 5% and up to 50% of cases, and include diarrhea, nausea, vomiting, anorexia, and abdominal pain. In about 5% of patients, gastrointestinal symptoms may occur prior to respiratory symptoms [20]. A meta-analysis of 1577 patients reported that among gastrointestinal symptoms, diarrhea was the most prevalent with 33.9%, followed by nausea with 12.5%, and vomiting with 11.5%. In this study, the presence of gastrointestinal symptoms was not associated with COVID-19 severity (Odds Ratio (OR) 1.16; 95% Confidence Interval (CI) 0.89–1.52), and only abdominal pain was associated with a more severe disease (OR 2.83; 95% CI 1.34–6.01; *p* = 0.007) [21]. By comparison, in another meta-analysis that included 6686 patients, the presence of gastrointestinal symptoms was associated with a higher risk of acute respiratory distress syndrome (OR 2.85, 95% CI 1.17–7.48), and abdominal pain was associated with greater disease severity (OR 7.10; 95% CI 1.93–26.07) [22].

A systematic review and meta-analysis involving 4682 patients reported that the most significant gastrointestinal symptoms were anorexia (17%; 95% CI 0.06–0.27) and diarrhea (0.08; 95% CI 0.06–0.11), and patients with severe disease were more likely to have diarrhea, anorexia, and abdominal pain [23]. Although the clinical effect of gastrointestinal symptoms on COVID-19 outcomes is mixed, a prospective study in 244 patients reported that diarrhea was not associated with mortality (0% vs. 7.7%, *p* = 0.036) and overall gastrointestinal symptoms were negatively associated with moderate to severe disease (*p* = 0.004) [24].

In a study evaluating the dynamics of fecal RNA shedding in 113 patients, fecal SARS-CoV-2 RNA was detected in 49.2% (95% CI 38.2–60.3%) of patients within the first seven days after diagnosis, and 3.8% (95% CI 2.0–7.3%) of individuals shed for up to seven months [25]. Although it has been postulated that shedding of the virus correlates with gastrointestinal and nongastrointestinal symptoms, no association with long COVID-19 symptoms has been found [26].

Complex gastrointestinal complications of COVID-19 have been reported in severe disease, including ileus, hepatic necrosis, acalculous cholecystitis, and bowel ischemia. A systematic review that included 22 studies reported that 29% of patients presented with arterial mesenteric thromboembolism and 19.3% with portal venous thrombosis, requiring laparotomy and bowel resection in 64.5% with an overall mortality of 38.7% [27]. Although rare, ischemic gastrointestinal complications of COVID-19 can be fatal. Figure 1 highlights the relationship between COVID-19 and gastrointestinal manifestations and complications.

## 4. COVID-19 and the Liver

At the beginning of the pandemic, the behavior of COVID-19 was expected to be similar to that of the SARS pandemic of 2003, where liver damage was reported in up to 60% of patients [31]. Early publications reported aberrations in liver tests, suggesting a possible mechanism of liver damage different from that in SARS-CoV-2. Abnormal liver function tests are frequently observed, with a prevalence as high as 76.3% and associated with more severe disease with abnormal alanine aminotransferase (ALT) (OR 1.89; 95% CI 1.30–2.76; *p* = 0.009) and abnormal aspartate aminotransferase (AST) (OR 3.08; 95% CI 2.14–4.42; *p* < 0.0001) [22].

The pattern of liver damage has been associated with greater severity of the disease with an OR of 2.73 (95% CI 1.19–6.3) for hepatocellular damage and an OR of 4.44 (95% CI 1.93–10.23) for mixed damage [32]. A meta-analysis that included 12,882 patients reported AST elevation in 41.1% and ALT elevation in 29.1% of cases; acute liver damage was reported in 26.5% of cases, which was associated with worse outcome (OR 1.68; 95% CI 1.04–2.70; *p* = 0.03) [33]. Abnormal liver tests in COVID-19 may be independent of the presence of pre-existing liver disease [34], which suggests a direct effect of the virus on the liver, among other mechanisms that are discussed below. In most cases, these alterations are transient and nonsignificant [19].

### 4.1. Proposed Mechanisms of Liver Injury

The liver is the only organ with double blood supply (arterial and portal) and harvests the largest reserve of macrophages, playing a crucial role in the immune response to SARS-CoV-2 through hepatic stellar cells. In addition, endothelial cells in liver sinusoids register and activate the immune response through Toll-like receptors [35].

The interaction between SARS-CoV-2 and the liver is under active investigation because ACE2 receptors are not expressed in Kupffer cells, hepatocytes, or endothelium of the hepatic sinusoids [7]. By contrast, ACE2 receptors are expressed in vascular endothelium and in cholangiocytes, almost in the same proportion as in the type II pneumocytes [31]. The endothelium of bile ducts is more susceptible to SARS-CoV-2, therefore the upregulation of ACE2 receptors favors internalization of the virus and results in liver damage due to compensatory proliferation of hepatocytes [19]. Interestingly, the expression of ACE2 receptors has been reported in fatty liver animal models [36] and in regeneration nodules in liver cirrhosis [35], giving rise to the possibility of greater susceptibility to liver damage in such patients.

Currently, there is no consensus on the exact mechanism of liver damage in COVID-19. However, several hypotheses have been postulated: (1) direct cytopathic damage, (2) systemic inflammatory response with immune-mediated collateral damage, (3) hypoxia and liver ischemia, as in hypoxic hepatitis, (4) acute-on-chronic liver failure, and (5) drug-induced and/or herbal-induced liver injury [37].

Direct cytopathic damage has not been demonstrated and in a study of postmortem liver biopsies, only moderate lymphocytic lobular infiltrate and centrilobular sinusoidal dilatation were reported, findings that the authors attributed to the patient’s previous comorbidities [38]. In another postmortem study, no viral inclusions were found in the liver parenchyma [39].

The terms “cytokine release syndrome” and “cytokine storm” have been described in the medical literature since 1992 and refer to the role of a devastating effect of immune dysregulation, characterized by constitutional symptoms, systemic inflammation, and multiorgan dysfunction that can lead to multiorgan failure and death [40]. The cytokine storm in COVID-19 is characterized by high circulating levels of interleukin-1β, interferon-γ, and monocyte chemoattractant protein-1 [35], which upregulates the expression of ACE2 and TMPRSS2 receptors [6].

Hypoxic hepatitis refers to the massive, rapid rise in serum aminotransferases as a result of reduced oxygen delivery to the liver, which most common in cardiac failure, septic shock, and respiratory failure, but may occur in the absence of hypotension or a shock state in about 50% of cases [41]. In COVID-19, hypoxic hepatitis may be secondary to septic shock, COVID-related myocarditis (as cardiogenic shock), and ventilator complications [42]. In a series of 40 patients who died of complications of COVID-19, congestion and centrilobular ischemic necrosis were found in 78% and 40% of cases, respectively [43]. Hypoxic hepatitis may be considered among the differential diagnoses of liver injury in patients with COVID-19.

Acute-on-chronic liver failure (ACLF) refers to an acute decompensation in patients with chronic liver disease that is associated with a high risk of short-term mortality [44]. This syndrome is characterized by intense systemic inflammation, a close precipitating event, and single or multiple organ failure [45]. The mortality burden of ACLF in wait-listed patients is high, with prompt liver transplantation required in survivors. It has been hypothesized that patients with cirrhosis and ACLF have an increased risk of developing severe COVID-19 because of immune dysregulation (or immune paralysis). In cirrhosis, immune dysregulation is responsible for 30% of the mortality and is characterized by increases in anti-inflammatory cytokines, suppression of proinflammatory cytokines, increased gut permeability, reduced intestinal transit, and altered intestinal microbiota, which increases the risk of bacterial translocation and endotoxemia [46]. In a study that included 2460 patients, 35% met the definition of ACLF from the European Association for the Study of the Liver (EASL)-Chronic Liver Failure Consortium and exhibited prolonged hospital stay (14.7 ± 17.3 days vs. 5.4 ± 5.3 days, *p* = 0.004), severe COVID-19 (25% vs. 3%, *p* = 0.03), need for intensive care unit (45% vs. 11%, *p* = 0.003), and higher mortality (30% vs. 5%, *p* = 0.01) than patients without ACLF [47].

Drug-induced liver injury (DILI) and herb-induced liver injury (HILI) are defined as liver dysfunction and/or abnormalities in liver function tests secondary to the use of medications, herbs, or xenobiotics within the reasonable exclusion of other etiologies [48]. Many drugs have been used to treat patients with COVID-19, including antiviral agents (e.g., lopinavir, ritonavir, remdesivir, darunavir, umifenovir, and favipiravir), antibiotics (e.g., azithromycin), antimalarials (e.g., chloroquine and hydroxychloroquine), monoclonal antibodies (e.g., tocilizumab), JAK inhibitors (e.g., baricitinib), tyrosin kinase inhibitors (e.g., imatinib) [49,50], complementary alternative medicine (e.g., chlorine dioxide and Ayurvedic Kadha), and home remedies (*Allium sativum*) [51], many of which have been associated with hepatotoxicity alone or in combination as compassionate treatment. In this context, the Réseau d’Étude Francophone de l’Hépatotoxicité des Produits de Santé, a European study network focused on DILI, reported four cases of lopinavir/ritonavir suspected hepatotoxicity [52]. In clinical practice, polypharmacy is not uncommon, and physicians must be aware of DILI and HILI as potential causes of liver injury in patients with COVID-19. The same work-up and recommendations for patients without COVID-19 should be started upon suspicion of polypharmacy and DILI, especially discontinuation of the offending drug.

The exact mechanism of liver damage in COVID-19 is complex, challenging, and multifactorial in nature.

### 4.2. Implications in Fatty Liver Disease

Information concerning the association of hepatic steatosis and fibrosis with COVID-19 outcomes is limited. A more severe illness and worse outcomes are expected because a higher expression of ACE2 receptors has been found in hepatocytes of animal models of fatty liver [36].

A retrospective study that included 202 patients with fatty liver assessed by a hepatic steatosis index (HSI) > 36 points and/or confirmation by liver ultrasound reported a higher risk of progression of COVID-19 (6.6% vs. 44.7%, *p* < 0.00001) and longer shedding time (17.5 days vs. 12.1 days, *p* < 0.00001) in comparison with patients without fatty liver disease [53].

Obesity and metabolic syndrome are common risk factors for metabolic associated fatty liver disease (MAFLD) and in many cases coexist with COVID-19 in an alarming way. A prospective study of 214 patients with COVID-19 reported a 30.8% prevalence of MAFLD, of whom 68.2% had obesity associated with greater severity of disease (OR 6.32; 95% CI 1.16–34.54, *p* = 0.033) [54].

Establishing the risk of MAFLD through noninvasive predictive models at the time of hospitalization for COVID-19 may result in an overestimation of the prevalence of MAFLD because biomarkers (e.g., transaminases) used in the models (e.g., HSI, NAFLD-FS) may be altered by COVID-19. In addition, imaging techniques such as ultrasound have an overall sensitivity of 84.8% (95% CI 79.5–88.9) and a specificity of 93.6% (95% CI 87.2–97.0) for the detection of moderate to severe fatty liver [55], which could be considered as a reliable bedside diagnostic tool for fatty liver and for excluding other causes of abnormal liver tests.

In a systematic review and meta-analysis including 16 observational studies and 1746 MAFLD patients, the prevalence of COVID-19 was 29% (95% CI 0.19–0.40, *p* = 0.04) and was associated with increased severity (OR 3.07; 95% CI 2.30–4.09) and risk of ICU admission (OR 1.46; 95% CI 1.12–1.91, *p* = 0.28) but not associated with mortality (OR 1.45; 95% CI 0.74–2.87, *p* > 0.05) [56]. Patients with MAFLD seem to be at higher risk of developing complications related to COVID-19, but further research is needed.

### 4.3. Implications in Liver Cirrhosis

Liver cirrhosis is estimated to affect 4.5–9% of the world’s population [57], and a high proportion of patients with cirrhosis are expected to be infected with COVID-19. A multicenter study of 160 patients reported a prevalence of advanced fibrosis of 28.1% assessed with FIB-4 ≥ 2.67, which was associated with a higher risk for requiring intensive care (OR 3.41; 95% CI 1.30–8.92) [58].

The two most important international registries of patients with chronic liver disease and COVID-19 are COVID-Hep.net (University of Oxford and EASL) and SECURE-Cirrhosis Registry (University of North Carolina at Chapel Hill). In the first report, which included 745 patients (386 with cirrhosis and 359 controls), a mortality of 32% was found in patients with cirrhosis, which increased according to liver disease severity to 35% (OR 4.14; 95% CI 1.03–3.52) in Child–Pugh B and 51% (OR 9.32; 95% CI 4.80–18.08) in Child–Pugh C. Acute decompensation was observed in 46% of cases, of which 21% had no respiratory symptoms and 50% presented as ACLF [59].

An additional concern during the COVID-19 pandemic is the role of immunosuppression in patients with autoimmune liver diseases due to the increased risk of respiratory tract infections. However, a greater severity of infection has not yet been demonstrated in this group of patients. In this regard, recommendations for the approach to this group of patients are summarized in Table 2 [60].

### 4.4. Implications in Liver Transplantation

Since 1980, transplant programs around the world have responded to the pandemics of HIV, SARS-CoV, East Nile Virus, Influenza A/H1N1, Zika, and Ebola, maintaining their operation with the evaluation of the transmission to the donor, the severity of the disease in the recipient, and the risk of transmission to health personnel [61]. Unlike other solid organ transplant programs where alternatives or bridging therapies exist, such as hemodialysis, cardiac assist devices, and extracorporeal membrane oxygenation, in the case of patients with acute liver failure or end-stage liver disease, liver transplantation is the only treatment alternative. Currently, liver transplantation programs are among the most vulnerable around the world.

During the COVID-19 pandemic, the availability of intensive care beds has been vital, and this need is shared by transplant programs whose patients require specialized postoperative care. Therefore, a staggered-phase approach has been proposed with a decrease in the activity of transplant programs according to the tolerance of transplant risk, hospital capacity, and pandemic activity in the locality [61]. In a health system completely overwhelmed by the care of COVID-19 patients, the transplant program must be reduced by 100%.

Early experience at the beginning of the COVID-19 pandemic has been reported in a prospective nationwide study containing 111 liver transplant patients with COVID-19. At a median of 23 days, up to 86.5% of patients were hospitalized, 19.8% required intubation, 10.88% were admitted to the intensive care unit, and 18% died [62]. Currently, there is no consensus regarding the time of liver transplant in patients with COVID-19. Even so, successful liver transplants have been reported in patients with asymptomatic COVID-19 [63].

Recently, a study involving 792 patients from the EASL-COVID-Hep network and 283 patients from the UK OCTAVE study were compared with 93 healthy controls from the UK PITCH consortium. The study reported that liver transplant recipients had reduced anti-S Ig titer following two doses of BNT162b2 or ChAdOx1 vaccines [64]. It is still unknown why liver transplant patients failed to generate a response following vaccination.

### 4.5. COVID-19 and Inflammatory Bowel Disease

Patients with inflammatory bowel disease (IBD) share a similar inflammatory cytokine profile with acute exacerbation and “cytokine storm”, and such patients could benefit from treatment with interleukin-1 or interleukin-6 antagonists [65].

Two forms of the ACE2 receptor have been described, a soluble one that lacks a transmembrane domain and circulates in small amounts in the bloodstream and a complete one made up of an extracellular domain and a transmembrane domain and is responsible for the internalization of the virus into the host cell [66]. The soluble ACE2 receptor is upregulated in patients with IBD, and in vitro studies have shown that it can prevent virus binding to the transmembrane ACE2 receptor by acting as a competitive inhibitor [65].

In a multicenter study from the SECURE-IBD database (University of North Carolina at Chapel Hill), 232 patients with IBD (101 with Crohn’s disease, 93 with ulcerative colitis, and 38 with indeterminate colitis) were compared with 19,776 controls, and although there was no difference between the groups regarding the severity of the infection, the risk of severe COVID-19 was higher in patients under treatment with steroids (OR 1.60, 95% CI 1.01–2.57; *p* = 0.04) [67]. From this database, 209 patients under 18 years of age were analyzed and hospitalization was required in 7% with 1% requiring mechanical ventilation; no mortality was reported. Interestingly, patients receiving anti-TNF monotherapy had a lower rate of hospitalization (7% vs. 51%, *p* < 0.01) [68].

A meta-analysis containing 24 studies reported a pooled incidence rate of COVID-19 of 4.02 per 1000 persons with IBD (95% CI 1.44–11.17, *I*^2^ = 98%) with a pooled relative risk of acquiring COVID-19 no different from the general population (0.47; 95% CI 0.18–1.26, *I*^2^ = 89%) nor type of IBD (1.03, 95% CI 0.62–1.71, *I*^2^ = 0); pooled mortality was 4.27% [69]. As reported in previous studies, the relative risk of hospitalization, intensive care unit admission, and mortality was lower for patients on biologics but higher for those taking steroids or 5-aminosalicylates. In a prospective study including 5457 patients with IBD (I-CARE project), 4.3% reported COVID-19 with 0.2% severe cases and no COVID-19-related mortality [70]. Currently, there is no evidence that IBD is associated with a higher risk of COVID-19 infection or worse outcomes.

Similar to autoimmune liver diseases, IBD relapse may occur with inappropriate discontinuation of corticosteroids, antimetabolites, immunomodulators, and/or biologics [71]. Therefore, discussion with IBD experts is warranted in the treatment and follow-up of these patients.

### 4.6. COVID-19 and the Pancreas

Because COVID-19 manifests as a multisystemic disease, pancreatic involvement is expected, although limited evidence is available. Multiple viruses affect the pancreas, including hepatotropic viruses (hepatitis A, B, and E, Epstein–Barr, Coxsackie, Cytomegalovirus, and herpes zoster), as well as HIV, mumps, measles, and varicella zoster [72].

Higher levels of ACE2 messenger RNA have been found in the pancreas than in the lungs, a finding that could explain the pancreatic damage with infection [73]. In addition, obesity increases visceral adipose tissue, including intrapancreatic fat, in which the content of unsaturated fatty acids from triglycerides in adipocytes is released by hydrolysis and perpetuates fat necrosis in cases of acute pancreatitis that ultimately results in a “cytokine storm” and multi-organ failure, a similar scenario to that observed in COVID-19 [72].

Although the presence of SARS-CoV-2 in the pancreas has not been demonstrated in necropsy studies of patients with COVID-19, in 2004 SARS-CoV was detected by murine monoclonal antibodies in four patients with SARS [74]. Because of the similarity in viral structure and pathogenesis of SARS-CoV with SARS-CoV-2, a similar distribution can be expected in tissues and organs between the two viruses.

Abnormal pancreatic enzyme levels have been described in 8.5% to 17.3% of COVID-19 cases. Nonetheless, only 0.76% of cases met the Atlanta criteria for acute pancreatitis [72]. In a multicenter retrospective study of 71 hospitalized patients with COVID-19, only 12.1% presented with hyperlipasemia and 2.8% had a report of serum lipase levels at least three times the upper limit of normal, which was not associated with worse outcomes and no patient met the criteria for acute pancreatitis [75]. In patients with severe COVID-19, a higher prevalence of elevated pancreatic enzymes has been reported, and in a retrospective study of 1003 patients, 16.8% presented with lipase more than three times the upper limit of normal, which was associated with a higher rate of admission to intensive care (OR 8.93; 95% CI 2.43–38.5, *p* < 0.002) and intubation (OR 12.5; 95% CI 2.95–68.4, *p* < 0.002) when reported in the range of 81 to 701 IU/L [76].

Most cases of acute pancreatitis have been documented in patients with severe COVID-19, even without respiratory symptoms at the onset of the disease [72]. In a multicenter retrospective study of 48,012 hospitalized patients, only 0.39% met acute pancreatitis criteria, of whom 17% had COVID-19, and this combination was associated with a higher rate of intubation (OR 5.65; 95% CI 1.49–21.52, *p* = 0.01) and length of hospital stay (OR 3.22; 95% CI 1.34–7.75, *p* = 0.009) compared with patients without COVID-19 [77].

Finally, treatments used for COVID-19 can precipitate acute pancreatitis both directly (e.g., steroids and baricitinib) and indirectly (e.g., hypertriglyceridemia from tocilizumab and lopinavir/ritonavir) [72]. Therefore, although acute pancreatitis is not the most frequent manifestation of COVID-19, it should be considered within the differential diagnosis of patients with gastrointestinal symptoms, specifically abdominal pain.

As expected, in a study comparing admissions for acute pancreatitis (baseline group) with admissions during the same period in 2020 (pandemic group), patients in the pandemic group were more likely to present with systemic inflammatory response syndrome (40% vs. 25%, *p* < 0.01) and pancreatic necrosis (14% vs. 10%, *p* = 0.03), reflecting an avoidance of hospitalization for milder cases [78].

## 5. Conclusions

The role of the gastrointestinal tract in terms of presentation, progression, and outcomes in COVID-19 infection is becoming increasingly more important. Gastrointestinal symptoms and liver damage are common and can be associated with worse outcomes. Patients with cirrhosis are at increased risk of severe COVID-19. Patients with IBD are not at higher risk of complications, except for those on steroids. Immunosuppressive treatment should be continued and adjusted according to the clinical scenario of each individual case. Evidence of pancreatic involvement in patients with COVID-19 is still scarce.

## Figures and Tables

**Figure 1 tropicalmed-07-00187-f001:**
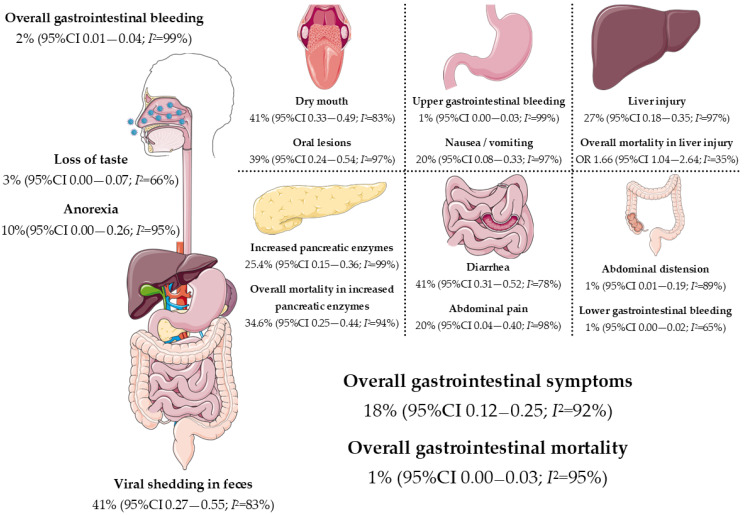
Overview of COVID-19 gastrointestinal, hepatic, and pancreatic manifestations (Adapted from references [14,28,29,30]).

**Table 1 tropicalmed-07-00187-t001:** Basic and translational implications of COVID-19.

Study	Hypothesis	Design	Results	Implications
Jiao [15]	The gastrointestinal tract could play a central role in the pathogenesis of COVID-19	Infection of Rhesus monkeys with an intragastric or intranasal challenge with SARS-CoV-2	Both intranasal and intragastric inoculation caused pneumonia and gastrointestinal dysfunction	Possible connections through inflammatory cytokines
Wang [16]	SARS-CoV-2 could be potentially transmitted other than through the respiratory tract	Biodistribution of SARS-CoV-2 among different tissues of inpatients	SARS-CoV-2 detected in respiratory tissue, feces, and blood but not in urine	Transmission of the virus through extra-respiratory routes (feces) could explain the rapid spread
Irham [17]	Individual expression of *TMPRSS2* may influence SARS-CoV-2 susceptibility	Multiple large genome databases (GTEx portal, SNP nexus, Ensembl genome project)	Four variants (rs464397, rs469390, rs2070788, and rs383510) affect expression of TMPRSS2 in lung tissue	Higher frequency of upregulating variants in European and American populations
Cao [18]	ACE2 variants could reduce the binding of S protein in SARS-CoV-2	Analysis of variants of ACE2 gene and allele frequencies in ChinaMAP and 1 KgP databases	Singleton truncating variant of ACE2 (Gln300X) and higher allele frequency in China of the SNP rs2285666	Lack of natural resistant mutations for coronavirus S protein binding

**Table 2 tropicalmed-07-00187-t002:** Recommendations for patients with autoimmune liver disease and COVID-19 (adapted from Lleo [60]).

Summary of Recommendations
Organize independent access to health services to avoid contact with COVID-19-positive patients. Limit invasive screening procedures to only emergency interventions (e.g., endoscopy). Initiate immunosuppressive treatment at standard doses for the treatment of exacerbation of autoimmune hepatitis. Coordinate care with the transplant committee in case of acute liver failure. Reduce immunosuppression in case of infection, especially antimetabolites in patients with lymphopenia.

## Data Availability

Not applicable.

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
