# Peer review of "COVID-19: Current Status in Gastrointestinal, Hepatic, and Pancreatic Diseases—A Concise Review"

_tropicalmed, 2022, doi:10.3390/tropicalmed7080187_

Round 1
Reviewer 1 Report
By a review, the authors claimed that the gastrointestinal symptoms might be the presentation of COVID infection. It should be more emphasized during the pandemic of COVID-19.Manuscript entitled "COVID-19: Current Status and in Gastrointestinal Diseases".
This work is potential of interest and could be acceptable for publication pending minor revision as followings:
1. The authors should summarize their thoughts in a figure to make it more clear.
2. Currently, the article contains less basic and translational ideas that should be added.
Author Response
Thank you very much for your review.
- A figure was designed (attached)
- A table was written for basic and translational ideas
Additionaly, article title was rewritten and currently the draft is under style correction.

Reviewer 2 Report
In the manuscript entitled “Covid-19: Current status and in Gastrointestinal Diseases” Aquino-Matus et al. have discussed a potential existence of gastrointestinal disorders due to Covid-19 infection. Authors have correlated the expression of Ace-2 in different types of cells across the organs liver, pancreas and intestine etc. with potential infection of Covid-19 however reports showing this link is very limited. Though there is not enough research to establish link between Covid-19 infection and intestinal, liver and pancreatic diseases, authors have managed to propose an idea that SARS-CoV2 infection could potentially induce diseases or worsen underline diseases of these organs. Overall, this article brings a new direction of thinking on SARS-CoV2 infection. I have few suggestions though.
1. Current title indicates that the article is on GI diseases however, authors have addressed the liver and pancreatic issues as well. So, my suggestion would be to change the title accordingly.
2. Take care of typos and grammars across the manuscript, including one in the title.
Author Response
Thank you very much for your review
- Title has been rewritten to:
COVID-19: Current Status in Gastrointestinal, Hepatic and Pancreatic Diseases. A concise review.
- Currently the final draft is under style correction.
Additionally, a figure and a table were included (attached).

Reviewer 3 Report
The authors comprehensively reviewed the relationship between COVID-19 and GI diseases. Most patients have gastrointestinal symptoms and/or abnormal liver function tests. Furthermore, they are associated with worse outcomes of COVID-19. However, the implications of COVID-19 in different GI diseases are still complicated. Therefore, we need table to summary these findings to make it easier to read.
Author Response
Thank you very much for your review:
- A figure was designed to clear the article findigns
Additionally, a table was written and currently the final draft is under style correction.
